# Serum Albumin: A Multifaced Enzyme

**DOI:** 10.3390/ijms221810086

**Published:** 2021-09-18

**Authors:** Giovanna De Simone, Alessandra di Masi, Paolo Ascenzi

**Affiliations:** 1Dipartimento di Scienze, Università degli Studi Roma Tre, Viale Marconi 446, 00146 Roma, Italy; alessandra.dimasi@uniroma3.it; 2Laboratorio Interdipartimentale di Microscopia Elettronica, Università degli Studi Roma Tre, Via della Vasca Navale 79, 00146 Roma, Italy; ascenzi@uniroma3.it; 3Accademia Nazionale dei Lincei, Via della Lungara 10, 00165 Roma, Italy

**Keywords:** human serum albumin, aldolase activity, anti-oxidant activity, enzymatic properties, enolase activity, esterase activity, glucuronidase activity, human serum heme-albumin, peroxidase activity

## Abstract

Human serum albumin (HSA) is the most abundant protein in plasma, contributing actively to oncotic pressure maintenance and fluid distribution between body compartments. HSA acts as the main carrier of fatty acids, recognizes metal ions, affects pharmacokinetics of many drugs, provides the metabolic modification of some ligands, renders potential toxins harmless, accounts for most of the anti-oxidant capacity of human plasma, and displays esterase, enolase, glucuronidase, and peroxidase (pseudo)-enzymatic activities. HSA-based catalysis is physiologically relevant, affecting the metabolism of endogenous and exogenous compounds including proteins, lipids, cholesterol, reactive oxygen species (ROS), and drugs. Catalytic properties of HSA are modulated by allosteric effectors, competitive inhibitors, chemical modifications, pathological conditions, and aging. HSA displays anti-oxidant properties and is critical for plasma detoxification from toxic agents and for pro-drugs activation. The enzymatic properties of HSA can be also exploited by chemical industries as a scaffold to produce libraries of catalysts with improved proficiency and stereoselectivity for water decontamination from poisonous agents and environmental contaminants, in the so called “green chemistry” field. Here, an overview of the intrinsic and metal dependent (pseudo-)enzymatic properties of HSA is reported to highlight the roles played by this multifaced protein.

## 1. Introduction

Human serum albumin (HSA) is the most abundant protein in plasma, representing the main determinant of the oncotic pressure and of fluid distribution between body compartments [1,2,3,4,5,6]. Moreover, HSA is the main carrier of endogenous and exogenous ligands, including fatty acids (FAs), nucleic acids, hormones, metals, toxins, and drugs, accounts for most of the pro- and anti-oxidant capacity of plasma, and displays (pseudo-)enzymatic properties [4,7,8,9,10,11,12,13]. Furthermore, HSA plays a pivotal role in heme scavenging acting as a depot and a carrier of the macrocycle; in fact, HSA transfers the macrocycle from high- and low- density lipoproteins (i.e., HDL and LDL, respectively) to hemopexin. Of note, the HSA-heme-Fe complex displays globin-like catalytic properties including peroxynitrite scavenging and catalase and peroxidase activities [4,14,15,16,17,18].

HSA is a valued biomarker of many diseases, (i.e., cancer, rheumatoid arthritis, ischemia, obesity, and diabetes) [19,20,21,22] and finds clinical application in the treatment of several pathologies including shock, trauma, hemorrhage, acute respiratory distress syndrome, hemodialysis, acute liver failure, chronic liver disease, and hypoalbuminemia [4,22,23,24,25,26,27,28,29]. In the last years, it is emerging a novel role of HSA in human innate immunity [9,12,30,31]. Indeed, HSA acts as a self-defense agent toward *Clostridium difficile* and *Streptococcus pyogenes*, as well as against *Candida albicans* infection by inactivating toxins [9,12,30,31,32]. Here, an overview of the intrinsic and metal dependent (pseudo-)enzymatic properties of HSA is reported to highlight the multiple roles played by this multifaced protein. 

## 2. Human Serum Albumin 

HSA is synthesized in the hepatocytes at the rate of ~0.7 mg/hour for gram of liver (i.e., at ~10–15 g/day), the serum concentration ranging from 35 to 70 g/L. The rate of HSA synthesis depends on the blood oncotic pressure, as the HSA concentration is detected by osmoreceptors in the hepatic interstitium. HSA biosynthesis is dependent upon hormone status, nutrition, and inflammation. However, the mechanisms underpinning HSA distribution throughout the body are openly debated [4,5,33,34,35].

### 2.1. HSA Structure

Circulating HSA is composed of 585 aminoacids (molecular weight: ~66 kDa); it contains a high number of Cys, Leu, Glu, and Lys residues, a low number of Met, Gly, and Ile residues, and only one Trp at position 214. The 35 Cys residues present in HSA form 17 disulfide bridges; Cys34 is the only unpaired amino acid residue exposing the free thiol to the solvent [1,4,22]. The predominance of ionized residues confers to HSA a high total charge that facilitates its solubility. Due to the prevalence of acidic aminoacids, at pH 7.0 HSA shows a net negative charge [4,22,36].

HSA is a monomeric globular protein composed by 67% α-helix without β-sheet, arranged in three domains (i.e., I, II, and III) encompassing amino acids 1–195, 196–383, and 384–585, respectively. Each domain includes 10 helices organized in subdomains A and B that are built from six and four α-helices, respectively, connected by a long loop [4,37,38]. The interface between subdomains IA and IB is linked to subdomain IIA by hydrogen bonds and hydrophobic interactions. This brings domains I and II to arrange themselves perpendicularly to each other in a T-shape arrangement [4,37,38]. Domain III contacts only subdomain IIB and organizes itself in a Y-shaped conformation with domains II and III. Moreover, domains I and III display few contacts, being separated by a crevice delimited by subdomains IA-IB-IIA on one side and subdomains IIB-IIIA-IIIB on the other [4,38,39,40,41]. The heart-shape positioning of domains I-II and II-III creates several sites able to bind endogenous and exogenous molecules [4,15,38,39,40,41,42,43,44] (Figure 1). 

### 2.2. HSA Oligomerization

Although native HSA is mostly monomeric, dimers, trimers, and higher order oligomers occur at high HSA concentrations [46]. HSA dimers occur at physiological pH values and HSA concentration; dimers exit the blood vessels into the extravascular fluids, which contain much lower levels of monomeric HSA, thus favoring dimers dissociation and shifting the equilibrium toward higher levels of monomers compared to dimers [47,48].

The self-oligomerization process is determined by non-covalent reversible interactions at or below physiological HSA concentrations. The formation of non-covalent dimers and oligomers takes place in response to high temperatures, extremes pH values, and other major stresses [46,47,49,50,51,52,53,54]. The interface of non-covalent dimers involves subdomains IA (hosting the Cys34 residue) and IB of two HSA molecules. Remarkably, also covalent dimers have been described, in which a disulfide bond links the Cys34 residues of each HSA monomer [47,55]. Although crystal structures have not been determined, it has been reported that the disulfide dimer requires the molecular distortion of the monomers to bring the Cys34 residues close enough to form a direct disulfide bond [47]. Some non-covalent and covalent dimers are irreversible, insoluble, and nearly all display a decrease of α-helixes and an increase of β-sheets. 

In the dimeric HSA, some binding sites of the monomeric form are no solvent accessible. Moreover, the structural changes following the dimer formation may induce allosteric changes affecting ligand binding and/or catalytic properties [47,48].

High levels of HSA dimers are present in the circulating blood of patients with chronic renal disease as compared with age-matched healthy individuals [56,57]. Besides, in the same patients also the concentration of products deriving from oxidative damage in blood have been reported [48,56,58], suggesting that the concentration of HSA dimers can be used as a biomarker of oxidative stress [48]. Moreover, in several diseases the Cys34 residue undergoes chemical modifications (see Section 3.7.1), this possibly affecting the monomer/dimer equilibrium and, in turn, the uptake and the release of ligands as well as the regulation of oncotic pressure [47]. 

Overall, dimers may offer the possibility to modulate some functions of HSA. Indeed, the HSA monomer/dimer equilibrium is sensitive to the binding of some ligands as well as to post-translational or chemical modifications of Cys34. So, a drug or a post-translational modification favoring dimer formation may in turn enhance the consequent binding and transport of specific ligands. In turn, the modulation of the monomer/dimer equilibrium of HSA may affect its physiological and therapeutical functions [47].

### 2.3. Ligand Binding Properties of HSA

FAs are the physiological ligands of HSA accommodating at nine sites named FA1 to FA9 [1,4,38,41]. Seven FA binding sites (i.e., FA1–FA7) are located in subdomains IB, IIIA, and IIIB while the FA8 and FA9 sites are positioned at the bottom and at the upper region of the gap between subdomains IA-IB-IIA on one side and subdomains IIB-IIIA-IIIB on the other [4,7,39,41,59,60] (Figure 1). FAs bind to the FA1-FA9 sites with different affinity: FA2, FA4, and FA5 are high-affinity pockets, FA1 and FA3 display a medium affinity, and FA6 and FA7 show a low affinity [41,61,62,63]. Of note, FA8 and FA9 sites display ligand occupancy only in the presence of short-chain FAs or saturating FA amounts, respectively; therefore, they are commonly considered as supplementary binding clefts [41,61] (Appendix A).

The FA1 site, also known as the heme binding pocket, is in a D-shaped cavity in the center of subdomain IB. It recognizes the carboxylate group of FAs by hydrogen-bonding to Arg117 and to a water molecule that in turn is coordinated by the side-chain hydroxyl group of Tyr161 and the carbonyl oxygen atom of Leu182. The tail of long-chain saturated FAs curls around the inside surface of the cavity so that the tip of the hydrophobic tail gradually approaches His146 at the lower end of the cavity opening [4,41]. The FA2 site, sited between subdomains IA and IIA, binds the FA carboxylate group by hydrogen bonding Tyr150, Arg257, and Ser287. The methylene FA tail extends linearly within the hydrophobic cavity generated by subdomains IA and IIA [4,41]. The FA3-FA4 cleft (also named Sudlow’s site II) is placed in the subdomain IIIA and binds two long-chain FAs. In the FA3 site, the carboxylate group of FAs establishes hydrogen bonds with Ser342 and Arg348 located within the IIB subdomain and with Arg485 placed in the IIIA subdomain. In the FA4 site, the FA carboxylate head-group is bound to Arg410, Tyr411, and Ser489 located in subdomain IIIA, while the FA tail locates into the hydrophobic tunnel of the subdomain IIIA [43,61,64]. The FA5 pocket, positioned in subdomain IIIB, hosts small ligands and binds a single FA molecule through Tyr401 and Lys525; the FA methylene tail enters the protein matrix tunnel [4,61]. The FA6 site is placed at the interface between subdomains IIA and IIB [4]. lacking residues that stabilize electrostatically the FA carboxylate, in turn it is recognized by the aliphatic portion of Arg209, Lys351, and Ser480 [61]. Remarkably, two molecules of the short-chain capric acid bind to FA6 in a linear tail-to-tail configuration [4,61,62]. The FA7 site (also named Sudlow’s site I), located within subdomain IIA, stabilizes the FA carboxylate by Arg257 [61,62]. The FA8 and FA9 sites. accommodating only short-chain FAs such as capric acid, are located at the base and at the top, respectively, of the crevice delimited by subdomains IA-IB-IIA on one side and subdomains IIB-IIIA-IIIB. In the FA8 site, the carboxylate head of capric acid binds to Lys195, Lys199, Arg218, and Ser454, placed on one side of the cavity, whereas the methylene tail of the FA bound to the FA6 pocket contributes to the formation of the hydrophobic end of the FA8 pocket [4,61,65]. A salt bridge connecting Glu187, located in the domain I, Lys432, placed in the domain III, contributes to stabilize FAs within the FA9 site (Appendix A).

HSA plays a relevant role as a drug (e.g., antibiotics, anti-coagulants, anti-neoplastics, anti-virals, anesthetics, anxiolytics, and non-steroidal anti-inflammatory drugs) carrier by influencing their pharmacokinetics and pharmacodynamics. The most relevant drug binding sites of HSA are the FA1, FA3-FA4, and FA7 pockets [4,41]. 

FA1 recognizes antibiotics, anti-neoplastics, anti-retrovirals, and non-steroidal anti-inflammatory drugs. Large antibiotics and anti-neoplastics bind to the FA1 site in the proximity of Tyr161 interacting with residues involved in heme and FAs recognition whereas smaller anti-retrovirals and non-steroidal anti-inflammatory drugs bind near His146 [4,15,41,66,67,68,69]. Ibuprofen is the prototypical ligand of the FA3-FA4 pocket that also recognizes anesthetics, anxiolytics, and the orphan drug 4-phenylbutazone [4,7,8,39,41,42,59,70,71,72]. The FA7 site is considered the primary binding site of HSA, the prototypical ligand being warfarin, a bulky heterocyclic anti-coagulant drug. In the presence of myristate, several drugs co-bind to the FA7 site. This suggests that myristate causes a conformational change(s) of the binding site resulting in the formation of three non-overlapping secondary sites able to recognize anesthetics, anticoagulants, antineoplastics, antiretrovirals, and non-steroidal anti-inflammatory drugs (NSAIDs) [13,41,73] (Appendix A).

## 3. HSA Enzymatic Properties

The enzymatic activity of HSA, regarding the hydrolysis of *p*-nitrophenyl esters, was reported for the first time in 1951 [74]. HSA shows esterase, enolase, glucuronidase, lipid peroxidase, and aldolase glutathione-linked thiol peroxidase activity (Table 1). Moreover, it shows DNA- and RNA-hydrolyzing activity, and anti-oxidant properties [4,75,76,77,78,79]. In addition, HSA displays heme-based globin-like (pseudo-)enzymatic properties including catalase, peroxidase, and NO/O_2_ detoxification actions [4,11,17,22].

### 3.1. Esterase (or Pseudo-Esterase) Activity of HSA

Although neither the physiological importance of the esterolytic activity of HSA nor its natural substrates have yet been clarified, HSA hydrolyzes different compounds, thus participating in several pharmaco- or toxicokinetic processes [77,79]. To date, is openly debated if the esterase-like activity of HSA can be classified as an enzymatic or rather pseudo-enzymatic action. Indeed, HSA undergoes irreversible acetylation upon reaction with *p*-nitrophenyl acetate, while enzymatic turnover requires both acetylation and deacetylation steps to obtain the final products and to restore the enzymatic activity [80,81].

HSA displays a (pseudo-)esterase activity with respect to: (i) α- and β-naphthyl acetate and *p*-nitrophenyl acetate [82,83,84], (ii) long and short-chain FA esters (e.g., *p*-nitrophenyl myristate (NphOMy), *p*-nitrophenyl esters of hexanoate (NphOHe), decanoate (NphODe) and *p*-nitrophenyl acetate (NphOAc)) [85,86], (iii) aspirin [87], (iv) ketoprofen glucuronide [76], (v) cyclophosphamide [88], (vi) esters of nicotinic acid [89], (vii) octanoylghrelin [90], (viii) nitroacetanilide [91], (ix) nitrotrifluoroacetanilide [92], and (x) organophosphorus compounds [93,94].

The esterase domains of HSA are located at the FA7 site where Lys199 plays a pivotal role for aspirin hydrolysis [1,4,75,77,95,96,97,98], and at the FA3-FA4 cleft (i.e., subdomain IIIA) where Tyr411 catalyzes the hydrolysis of long- and short-chain FA esters [4,86,95].

**Table 1 ijms-22-10086-t001:** Catalytic properties of HSA ^a^.

Enzymatic Activities	Substrates	Cofactors and/or Coenzyme	Catalytic Sites	Catalytic Residues	Recognition Residues	Kinetic Parameters		Ref.
						*K*_s_ or *K*_m_ (μM)	*k*_+2_ (s^−1^)	
Esterase activity	aspirin	n.n.	FA7	Lys199 His242 Arg257	Trp214 Arg218 Leu219 Arg222 Phe223 Leu238 Arg257 Leu260 Ile264 Ser287 Ile290 Ala291			[97]
Pseudo-esterase activity	chloropyriphosoxon	n.n.	FA3–FA4	Tyr411	Arg410	2.6 × 10^2 b^	6.6 × 10^‒4 b^	[99]
	ketoprofen glucuronide	n.n	FA3–FA4	Tyr411	Ser489 Leu491			[76]
	NphOAc	n.n.	FA3–FA4	Tyr411	Asn391 Leu407 Arg410 Lys414 Leu430 Leu453	(4.8 ± 0.5) × 10^−4 c^	(3.9 ± 0.4) × 10^−1 c^	[100]
	NphODe	n.n.	FA3–FA4	Tyr411	Asn391 Leu407 Arg410 Lys414 Leu430 Leu453	(2.3 ± 0.2) × 10^−5 c^	(8.1 ± 0.8) × 10^−4 c^	[86]
	NphOHe	n.n.	FA3–FA4	Tyr411	Asn391 Leu407 Arg410 Lys414 Leu430 Leu453	(2.9 ± 0.3) × 10^−5 c^	(2.8 ± 0.3) × 10^−3 c^	[86]
	NphOMy	n.n.	FA3-FA4	Tyr411	Asn391 Leu407 Arg410 Lys414 Leu430 Leu453	(2.6 ± 0.3) × 10^−5 c^	(1.6 ± 0.2) × 10^−4 c^	[101]
	*p*-nitrophenyl-*N*-methylcarbamate	n.n.	FA3–FA4	Tyr411		1.4 × 10^2 d^		[82]
	paraoxon	n.n.	FA3–FA4	Tyr411	Arg410	3.7 × 10^2^ ^b^	2.5 × 10^−5 b^	[99]
RNA-hydrolyzing activity	RNA	n.n.	FA7	Lys195 Lys199				[102]
		n.n.	FA3–FA4	Lys541 Lys545				[103]
Enolase activity	dihydro-testosterone	n.n.	NTS	His3				[104]
Glucuronidase activity	carprofen glucuronides	n.n.	FA7	Tyr411				[105]
	oxaprozin glucuronides	n.n.	FA3–FA4	Tyr411	Asn391 Leu407 Arg410 Lys414 Leu430 Leu453			[106,107]
Peroxidase activity	α-tocopherol	palmitoyl-CoA	FA3–FA4	Cys392 Cys438				[108]
	LDL	palmitoyl-CoA	FA3–FA4	Cys392 Cys438				[108]
Aldolase activity	aromatic aldehydes with acetone	n.n.	FA7	Lys199				[109]
Direct anti-oxidant activity	ROS (i.e., H_2_O_2_, O_2_^−^, HOCl)	Cu(I/II) Fe(II/III)	NTS MBS	Cys34				[22]
	RNS (i.e., ONOO^−^)	Cu(I/II) Fe(II/III)		Cys34				[22]
Indirect anti-oxidant activity	bilirubin	n.n.	FA1		Lys190 Lys240			[110]
	heme	n.n.	FA1		Tyr161 His142 Lys190			[111] [112]

^a^ FA, fatty acid; LDL, low density lipoproteins; MBS, metal binding site; n.n., not necessary; NphOAc, *p*-nitrophenyl acetate; NphODe, *p*-nitrophenyl decanoate; NphOHe, *p*-nitrophenyl hexanoate; NphOMy, *p*-nitrophenyl myristate; NTS, *N*-terminal site; RNS, reactive nitrogen species; ROS, reactive oxygens species. ^b^ pH 7.4 and 37.0 °C. ^c^ pH 7.5 and 22.0 °C. ^d^ pH 8.0 and 37.5 °C.

At the FA7 site, Lys199 acts as the nucleophile [77,113], His242 as the base, and a carbonyl group from the protein backbone stabilizes the His catalytic triad [77]. Lys199 hydrolyzes the substrate (e.g., acetylsalicylic acid (aspirin), trinitrobenzene-sulfonates, and penicillin) in two products, one of which is released while the other binds covalently to Lys199 [1,4,114,115]. The ability of Lys199 to attack the substrate is thought to be due to the proximity of Lys195. Indeed, the basic form of Lys199 residue is probably linked to the acid form of Lys195 by a network of H-bonding water molecules with a donor-acceptor character. The presence of these water bridges is important for the stabilization of the FA7 site configuration and/or for promoting a potential Lys195-Lys199 proton transfer process [1]. Furthermore, since Lys199 is located at the entry of the FA7 site, ligand binding to Lys199 inhibits the Lys199-dependent esterase activity [64,116]. Molecular docking simulations and quantum mechanics/molecular mechanics calculations conducted on the HSA:aspirin complex lead to speculate that Lys199, His242, and Arg257 may contribute to the (pseudo-)esterase activity of HSA [77].

The most catalytically efficient configuration of the FA7 active site requires that both Lys199 and aspirin are in the neutral form [77]. Aspirin stably acetylates HSA at Lys199 with the concomitant release of salicylate [98] and the modification of the HSA secondary structure [1,96,97,117] FAs modulates the affinity of aspirin and anti-coagulants for both BSA and HSA [4,118]. Indeed, both aspirin and salicylate bind to the myristate:HSA complex at the FA7 site with an orientation depending on the acetylation status of Lys199 [97]. In the absence of myristate, both the FA3-FA4 and FA7 sites are occupied by aspirin molecules [119]. In the presence of myristate, the salicylate molecule is sited only in the FA7 site [97]. The phenyl group of salicylates is positioned in the hydrophobic pocket paved by acetyl-Lys199, Leu219, Arg222, Phe223, Leu238, Leu260, Ile264, Ile290, and Ala291. In detail, Leu238, Leu260, Ser287, and Ile290 surround the carboxylate group of the salicylate, while acetyl-Lys199, His242, and Leu238 limit the hydroxyl group of salicylates. Acetyl-Lys199 is closed to Trp214, Arg218, Leu238, and His242 being hydrogen bonded to Arg218. In the myristate:HSA:aspirin complex, the acetyl-Lys199 side chain extends to position its amine group toward the hydroxyl group of salicylate [97]. The position of salicylate in the myristate:HSA:aspirin complex is different from that reported in the myristate:HSA:salicylate adduct. Indeed, the salicylate of the myristate:HSA:aspirin adduct is close to the entrance of the FA7 site and distant from the hydrophobic binding pocket formed by Leu238, Leu260, Ile264, Ile290, and Ala291. This difference results in a lower binding affinity of the hydrolysis product(s) of aspirin for acetyl-Lys199 compared to aspirin [97]. Interestingly, salicylate inhibits the (pseudo-)esterase activity of HSA. This might be due to the higher binding affinity of salicylate toward HSA compared with that of aspirin [120]. Indeed, salicylate forms more and/or closer contacts with HSA than with acetyl-Lys199 HSA [97] (Figure 2).

HSA also catalyzes the hydrolysis of long and short-chain FA esters NphOHe, NphODe, and NphOMy at Tyr411 [86]. The pH dependence of the kinetic parameters for the hydrolysis of NphOHe and NphODe reflects the acidic p*K*_a_-shift of the Tyr411 residue upon substrate binding from 8.9 in free HSA to 7.6 and 7.0 in the HSA:NphOHe and HSA:NphODe complexes, respectively. This could reflect the reduced solvent accessibility of Tyr411, which is in the FA3-FA4 cleft [43,61,64,86]. In addition, this renders more stable the negative charge on the phenoxyl O atom of Tyr411, which is hydrogen-bonded to the carbonyl O atom of *p*-nitrophenyl propionate [121]; this potententiates the Tyr411 nucleophilic role as the electron donor [86]. 

Diazepam, diflunisal, ibuprofen, 3-indoxyl sulfate, and propofol are competitive inhibitors of the HSA-catalyzed hydrolysis of NphOAc [101], NphOHe [86], NphODe [86], and NphOMy [86,101] impairing the accessibility of *p*-nitroplenyl esters to the Tyr411 catalytic center. Diazepam, diflunisal, ibuprofen, and 3-indoxyl-sulfate bind to the center of the FA3-FA4 cleft, with one O atom being hydrogen bonded to the Tyr411 OH group. On the other hand, the phenolic OH group of propofol establishes a hydrogen bond with the carbonyl oxygen atom of Leu430, belonging to the apolar region of the FA3-FA4 cleft. Propofol binds to the apolar region of the FA3-FA4 cleft through hydrogen bonding to the carbonyl O atom of Leu430. Moreover, propofol establishes with one of its two isopropyl groups several apolar contacts at one end of the pocket, while the other one, is solvent exposed at the cleft entrance contacting Asn391, Leu407, Arg410, and Tyr411. Furthermore, the aromatic ring of propofol is inserted between the Asn391 and Leu453 side chains. One of the two isopropyl groups of propofol is solvent exposed and makes close contacts with Asn391, Leu407, Arg410, and Tyr411 whereas the other establishes several apolar contacts at one end of the pocket [43,61,64,86].

Inter-lots variations in the HSA-catalyzed cleavage of aspirin and *p*-nitrophenyl acetate have been reported, suggesting that different commercial HSA preparations may affect the pharmacokinetic profiles of drugs [78]. Among others, these differences may be due to cholinesterase and sulfhydryl contaminations as well as to FAs content that may alter the structure of HSA catalytic sites [78,122].

Interestingly, a specific method to detect the HSA (pseudo-)esterase activity has been developed [123]. This method is based on a staining protocol based on the use of: (i) Fast Blue BB (0.12%) dye to stain the (pseudo-) esterase activity of HSA both on agarose and Native PAGE; (ii) 2-naphthyl acetate, an ester substrate of HSA, and (iii) neostigmine, a selective acetyl-cholinesterase inhibitor [123]. This method could find application in either HSA pharmaceutical preparation (e.g., to check adulteration of human milk with cow milk) [123,124,125] or in diagnostic detection (e.g., to highlight the (pseudo-)esterase activity of HSA in urine representing a sign of microalbuminuria) [123,126]. 

HSA has also been suggested to display the esterase activity towards nucleic acids. DNase activity has been hypothesized to occur at NTS and MBS since it appears to be metal-dependent [10,11] RNase activity has been attributed to Lys195-Lys199 and Lys541-Lys545 dyads since post-translation modifications of these residues (i.e., glycation, *N*-homocysteinylation, and *N*-phosphorylation) inhibit the RNA hydrolysis [10,102,103,127].

### 3.2. Enolase Activity of HSA 

HSA displays the enolase activity towards dihydro-testosterone converting it from the 3-keto to the 3-enol form [104]. The HSA-enolase activity decreases linearly with the protein concentration, has the optimum pH value at 9.2, and is inhibited by Ni(II) and Cu(II) binding to His3 as well as by oleic acid, cholesterol, and surface-active agents [4,104]. Polymeric HSA present in the malignant tissues displays a reduced enzymatic activity compared to the monomeric form that is exclusively present in benign breast specimens. This observation offers a possibility of reliable differentiation between benign and malignant breast tumors [96,104].

### 3.3. Glucuronidase Activity of HSA

Glucuronidation is a reaction of the phase II detoxification pathway that enhances the hydrophilicity of the parent drug by its biotransformation and elimination into urine and bile [4,94,128]. This crucial biological process, catalyzed by uridine di-phospho-glucuronosyl-transferases, consists in D-glucuronic acid binding to drugs bearing a hydroxyl, an amine, a thiol, or a carboxylic acid group [4,128]. Both in vivo and in vitro, glucuronide conjugates are intrinsically reactive molecules that can undergo a number of reactions; their hydrolysis, leading to the parent aglycon, is the predominating process [76,128,129,130]. Interestingly, HSA binds reversibly or irreversibly glucuronide conjugates decreasing their plasma levels [76,131,132] and facilitates their isomerization and stereo-selective hydrolysis [76,133,134,135]. Competition binding assays suggest that HSA glucuronidase activity lies both in Sudlow’s sites I (e.g., diflunisal glucuronide) or II (e.g., carprofen glucuronides, and oxaprozin glucuronides) depending on the chemical structure of the aglycone. Furthermore, Lys195 and Tyr411 are involved in the hydrolysis of glucuronide of *S*-carprofen, whereas other undetermined Lys residues are involved in the hydrolysis of glucuronide of *S*-carprofen diastereoisomer [134]. 

### 3.4. Lipid Peroxidase Activity of HSA

HSA exhibits a thioredoxin-dependent lipid peroxidase activity both at pH 8.0 and in the presence of palmitoyl-CoA at physiological pH [4,108,136]. In both cases, the enzymatic activity is promoted by a conformational change of HSA leading to the exposure of the Cys392–Cys438 catalytic bridge, which is responsible for the thioredoxin-dependent lipid peroxidase activity [108]. 

The chemical modification of Cys392 and Cys438 with the thiol-specific agent *N*-ethylmaleimide causes the loss of peroxidase activity. On the contrary, carboxyl group-modified HSA displays a 10-fold stronger lipid peroxidase activity than that of the native species [4,108,136]. These chemical modifications indicate that the redox state of HSA modulates the lipid peroxidase activity in vivo [108,136]. In vivo experiments suggest that: (i) analbuminemic rats are more susceptible to oxidative stress than normal rats and more easily develop tumors of the bladder, kidney, and stomach [108,137,138]; (ii) HSA prevents lipid peroxidation in rabbit spermatozoa protecting from loss of motility [108,139], (iii) HSA inhibits the peroxidation of erythrocyte membrane lipid, and (iv) persistent hypoalbuminemia reduces the serum antioxidant activity in chronic hemodialysis patients and may contribute to increased oxidative cell damage [108,140]. To date, the physiological function of HSA as a lipid peroxidase in extracellular fluids is still unsolved [4,108].

### 3.5. Aldolase Activity of HSA 

HSA catalyzes the aldol reaction of aromatic aldehydes with acetone [4,109]. The aldolase activity takes place in the FA7 site of has; Lys199, surrounded by a hydrophobic environment, catalyzes the aldol/retroaldol reaction by the enamine/Schiff base mechanism [4,109]. Lys199 is involved in the covalent binding of substrates and is responsible for the HSA ability to act as an enzyme-like catalyst in β-eliminations, in the decomposition of Meisenheimer adducts, and in the Kemp elimination [4,109]. HSA catalyzes reactions with a moderate and opposite enantio-preference possibly reflecting the different orientation assumed by the substrate in the active site. Enamine attacks the aldehyde carbonylic group Re face in HSA leading to the R product with 60% enantiomer excess [4,109]. The aldol reaction is one of the most important reaction in organic synthesis, allowing the formation of a C-C bond with the consequent generation of one or two stereocenters. With the aim of synthetizing chimeric proteins that combine binding and catalysis properties, a scaffold composed of the glutathione *S*-transferase fused to the HSA subdomain IIA has been engineered. This glutathione *S*-transferase-HSA chimeric protein retains both the binding properties of HSA and the aldolase activity representing a potential scaffold to produce a library of catalysts with improved proficiency and stereoselectivity for synthetic applications [109,141]. 

### 3.6. Glutathione-Linked Thiol Peroxidase Activity of HSA

HSA exhibits thiol-specific anti-oxidant properties that prevent the inactivation of glutamine synthetase activity and the peroxidation of lipid by a metal-catalyzed oxidation system. Indeed, HSA reduces H_2_O_2_ in the presence of reduced glutathione leading to the production of oxidized glutathione. Furthermore, the thiol-specific anti-oxidant properties of HSA are significantly activated by halide ions, especially chloride ion, and completely abolished by the reaction with *N*-ethylmaleimide and iodoacetate [4,142].

### 3.7. Anti-Oxidant Activity of HSA

Oxidative stress is a phenomenon caused by an imbalance between production and accumulation of reactive oxygen and nitrogen species (ROS and RNS, respectively) in cells and tissues and the ability of a biological system to detoxify these reactive products. ROS and RNS are toxic by-products of cellular metabolism with the potential to attack lipids, proteins, and DNA. High levels of ROS and RNS promote, among others, oxidative damage, inflammation, endothelial dysfunction, and renal tissue fibrosis [143]. ROS and RNS are mainly produced by mitochondria during cellular respiration, arachidonic acid metabolism, and inflammation [144]. HSA acts as a major and predominant anti-oxidant in plasma exerting more than 80% of the free radical-trapping activity of serum [4,22,145]. HSA detoxification from ROS relies mostly on glutathione-linked thiol peroxidase activity involving Met residues 87, 123, 298, 329, 446, and 548, Cys34, and metal ions [4,22,142,145,146] (Figure 3). The antioxidant activity of HSA reflects principally its metal-chelating properties. Indeed, surface-exposed Met residues of HSA can be oxidated to sulfoxide [147], which can be reverted to Met by methionine sulfoxide reductase or mild reductants [148]. Met sulfoxide may represent an endogenous anti-oxidant defense that protects proteins from irreversible oxidative modification [149]. 

HSA interacts with low-molecular-weight thiols also though the Cys residues 75, 90, 91, 101, 124, 200, 265, 392, 487, and 567 that form disulfide bridges [22,150,151]. According to the proposed mechanism for Cys thiolation, the free thiol group of Cys34 is thiolated as first with the consequent formation of the thiolate anion RS^−^ [22,151]. Then, RS^−^ attacks one of the 17 disulfide bonds of HSA triggering a cascade of radical reactions that stop when no more disulfide bonds are available on the protein surface for the thiolate anion. The partial reduction of the disulfide bonds increases the HSA hydrophobicity favoring protein aggregation and altering its functional properties [22,151].

Polyvalent Cu(I/II) and Fe(II/III) ions (either free or bound to heme) efficiently generate ROS after reaction with O_2_. Furthermore, Cu(I) and Fe(II) ions reacting with H_2_O_2_ lead to the formation of deleterious hydroxyl radicals by the Fenton reaction [4,22,152]. By binding Cu(I) and Fe(II) ions, HSA reduces their reactivity and prevents them from participating to free radical reactions with other blood components [22,153]. The main binding site of Cu(II) is represented by the *N*-terminal site (NTS) (i.e., Asp1-Ala2-His3), which displays a superoxide dismutase-like activity preventing ROS formation [22,154]. In vitro experiments suggest that HSA prevents neuronal death in murine cortical cell cultures exposed to H_2_O_2_/Cu(I)/ascorbic acid acting as an oxidant [155]. Recently, HSA has been suggested to trap the Cu(I) cation through the His pairs 67–247 (located in the metal binding site (MBS)) and 3–9 (at the *N*-terminus) [156]. HSA plays a pivotal anti-oxidant role in Fe(III)- and heme-Fe(III)-overload diseases [4,17,157,158,159]; however, Fe(III) and heme-Fe(III) recognition does not involve neither NTS or MBS [22].

HSA shows also indirect anti-oxidant activity by binding several compounds like polyunsaturated fatty acids [22,160], bilirubin, and homocysteine, and oxysterols. Bilirubin binding to HSA at subdomain IB inhibits lipid peroxidation, protects α-tocopherol from peroxyl radicals damage, and extends the survival of human ventricular myocytes against in situ-generated oxidative stress [4,160,161,162,163]. Remarkably, homocysteine trapping by HSA reduces the extent of homocysteine-dependent oxidative stress that causes vascular inflammation in atherosclerosis [164]. Indeed, an inverse relationship exists between the levels of oxidized HSA and risk of atherosclerosis [22,165]. HSA binds also oxysterols (i.e., biologically active oxidized forms of cholesterol or of its precursors) with an higher affinity compared to cholesterol [4,22,160,166], thus playing a key role in reducing oxysterols cytotoxic effects [160].

HSA has been proposed also to act as a carrier of reactive sulfur species (RSS), thus acquiring an anti-oxidant activity. In vitro studies indicate that the treatment of melanoma B16 cells with *S*-sulfhydrated HSA determines a significant anti-oxidant activity and inhibit melanin synthesis [22,167]. 

#### 3.7.1. Modulation of the HSA Anti-Oxidant Activity

During its long lifetime, HSA molecule makes about 15,000 passes through the circulation [1] undergoing damages that impairs its ligand binding and anti-oxidant properties [4,126]. The HSA anti-oxidant activity depends on: (i) the reactivity of the Cys34 residue, which in turn is affected by chemical modifications (e.g., oxidation and cysteinylation), (ii) post-translational modifications like glycation, (iii) FAs binding, and (iv) aging.

##### The Cys34 Residue

The Cys34 residue plays a relevant role in the anti-oxidant activity of HSA [22,146]. Cys34 acts as a scavenger of ROS (i.e., hydrogen peroxide, superoxide anion, and hypochlorous acid) and RNS (i.e., peroxynitrite) by oxidizing the -SH group to sulfenic acid (Cys34-SOH) [160,168]. In turn, sulfenic acid can be either irreversibly oxidized to sulfonic acid (Cys34-S(O)O^−^O^−^) or converted to disulfides (Cys34-S-S-R) upon reaction with low-molecular-weight blood plasma thiols (i.e., glutathione (GSH), homocysteine (HCys), cysteinylglycine (CysGly), glutamylcysteine and free cysteine) [151,169,170]. The Cys-S-S-Cys HSA dimer is the strongest oxidizing agent. Under pathological conditions, when the average of oxidazed Cys34 increases, HSA maintains a safe Cys-SH/Cys-S-S-Cys ratio [170]. The remaining Cys residues of HSA forming 17 disulfide bridges do not undergo cysteinylation even in the presence of high levels of free cysteine [170].

At physiological pH values, the Cys34 residue exists primarily as thiolate anion. Low molecular weight thiols present in plasma can reversibly oxidize 30–40 % of HSA at Cys34 [8]. Notably, protein homocysteinylation is one of the principal mediators of homocysteine toxicity as it induces structural and functional alterations in proteins and represent the pathological hallmark of cardiovascular and neurodegenerative disorders [171]. Levels of *S*-homocysteinylation can increase in the presence of hyperhomocysteinemia, a pathological condition in which the inherited or acquired alterations of HCys metabolism result in high plasma HCys concentrations (50–500 μmol/L) [172]. Besides, the HCys reactive product derived from the enzymatic conversion of HCys into the corresponding thioester, namely homocysteine thiolactone (HCys-T), can acylate the side-chains of Lys233 and Lys525 residues of HSA by a non-enzymatic nucleophilic reaction [173,174]. In turn, this leads to the *N*-homocysteinylation of HSA [175] and induces the formation of insoluble toxic amyloids [171,176]. Indeed, HSA *S*-homocysteinylation by HCys and *N*-homocysteinylation by HCys-T represent the most common post-translational modifications upon HSA secretion that modulate its binding and enzymatic properties [174]. HSA binding affinity for endogenous (e.g., bilirubin and tryptophan) and exogenous ligands (e.g., warfarin and diazepam) decreases as a function of Cys34 cysteinylation [177,178]. Indeed, levels of Cys34-cysteinylated HSA increase significantly in patients affected by diabetes mellitus and by diseases involving liver and kidneys [177].

The redox status of the thiol group of Cys34 allows to define three HSA isoforms [178,179]: (i) the free thiol identifies the reduced human mercaptalbumin (HMA); (ii) mixed disulfides with either Cys, CysGly, HCys, and GSH are typical of non-mercaptalbumin-1 (HNA-1); and (iii) the sulfinic or sulfonic acids characterize the non-mercaptalbumin-2 (HNA-2) [178,179]. HNA-1 is a reversible derivative whereas HNA-2 is irreversible. In healthy young people HMA is 70–80% of total HSA, HNA-1 is 20–30%, and HNA-2 is only 2–5% [178,179]. Increased HNA-1 levels in plasma are associated with a decreased HSA anti-oxidative activity [178]. HNA-1 and HNA-2 isoforms are related to the progression of inflammatory processes [178,180] whereas the concentration of HMA in the blood plasma is inversely correlated to atherosclerotic damages in the carotid artery of Japanese residents [165]. Thus, oxidation status of HSA represents an oxidative stress marker [178]. Notably, HSA oxidation represents a biomarker of Duchenne muscular dystrophy [181], Alzheimer and Parkinson disease [182,183], hyperparathyroidism [184], and acute ischemic stroke [185].

The COVID-19 pandemic is an ongoing global pandemic of coronavirus disease 2019 (COVID-19) caused by severe acute respiratory syndrome coronavirus 2 (SARS-CoV-2). It has been suggested that the cytokine storm observed in COVID-19 patients is caused by increased levels of HNA in serum. In turn, HNA increase may represent a predictor of mortality in elderly individuals as well as in patients affected with inflammatory diseases, cardiovascular disorders, and diabetes [178,186,187]. Notably, HSA oxidation is usually associated to a reduction of HSA plasma levels, a condition defines as hypoalbuminemia, and a higher mortality risk in COVID-19 patients [22,188].

The oxidation status of the Cys34 residues affects the binding affinities of HSA for lipid mediators. Indeed, while proatherosclerotic lipids (e.g., lysophosphatidylcholine, lysophosphatidic acid) have higher affinity for the oxidized isoforms of HSA, the antiatherosclerotic mediators derived from eicosapentaenoic and docosahexaenoic acid bind with higher affinity the reduced form of HSA [178].

##### HSA Glycation

The principal sites of HSA glycation are Arg410 and Lys525, whereas minor glycation sites are Arg10, Lys12, Lys51, Arg98, Arg114, Arg160, Lys199, Lys233, Lys276, Lys281, Lys317, Lys323, Arg428, Lys439, and Lys545. HSA glycation: (i) reduces the α-helical content by 10.4%, (ii) increases the intrinsic flexibility of domains and secondary structure elements, and (iii) exposes hydrophobic sites to the solvent. Under in vitro conditions, HSA glycation favors the formation of soluble HSA oligomers [189], molten globule-like [190] and amyloid-like [191] structures.

Data concerning the in vitro effects of glycation on the HSA anti-oxidant properties are still controversial, possibly because of the formation of: (i) different glycation intermediates, (ii) different types and concentrations of carbohydrate (e.g., glucose, methylglyoxal), and (iii) different experimental conditions [22,192,193].

In patients affected by diabetes mellitus, HSA glycation affects both ligand binding and anti-oxidant properties [146,147,148,149,150,151]. Glycated HSA shows a reduced ability to bind Cu(II) with respect to the non-glycated protein [147]. In turn, free Cu(II) induces LDL oxidation, probably by generating superoxide [4,146,147]. The 30–40% of diabetic patients develop nephropathies that require hemodialysis treatment; in turn, hemodialysis causes increased levels of chemically- and/or structurally-modified HSA with the consequent reduction of its anti-oxidant and binding properties [4,152]. Glycated HSA also displays a reduced capability to bind Fe(III) [4,149] and Trp, which is the only free amino acid transported by HSA [150]. The receptor-mediated recognition of advanced glycation end-products such as glycated HSA initiates the intracellular signaling and increases ROS formation in cells [153,154]. The HOCl-dependent carbonylation of Lys residues of glycated HSA is considered an advanced glycation end-product in hyperglycemia and inflammation [154]. In addition, advanced glycation end-products such as glycated HSA affect vascular endothelial NO synthase activity in rabbit aortas in vivo [155]. Glycated HSA shows a toxic effect on microglial cells associated with impairments in cellular proteolytic systems, thus suggesting the role of advanced glycation end-products in neurodegeneration [148,156,157,158]. Lastly, high levels of glycation impair HSA anti-oxidant properties, this possibly implying a predisposition to obstructive sleep apnea syndrome onset [4,151].

##### FAs Binding

FAs binding to HSA plays a crucial role in the regulation of anti-oxidant properties [22,194]. Indeed, upon FAs binding, Sudlow’s sites I and II undergo conformational changes as detected by fluorescence quantum yield increase of specific probes (i.e., dansylamide and dansylsarcosine for Sudlow’s site I and II, respectively). Moreover, FAs binding increases the steric availability of the Cys34 thiol group [22,194]. 

##### Aging

Oxidative stress increases with aging [187]. This process seems to be correlated to the plasma levels of HNA-1 and HNA-2. Moreover, cysteinylation and homocysteinylation of HSA are higher in the elderly than in the young [179,183,195,196].

## 4. Human Serum Heme-Albumin

HSA shows time-dependent heme-based catalytic properties, overall defined as “chronosteric effects” [17]. This reflects the pivotal role of HSA in the iron-macrocycle transfer from high- and low-density lipoproteins to hemopexin, thus acquiring globin-like reactivity [17,60,68]. After the CD91 receptor-mediated endocytosis, the hemopexin:heme complex entries into the hepatic parenchymal, releasing the heme that undergoes either degradation or recycling [17,60,68]. Under physiological conditions, the HSA-hemeFe(III) plasmatic level is about 1.0 × 10^−6^ M; in contrast, in patients with severe hematologic diseases the plasmatic levels of HSA-heme-Fe(III) increase, reaching a maximum of about 4.0 × 10^−5^ M [4,17,197,198]. Accordingly, HSA can be considered as the main depot of heme-Fe(III) [4,17,197,198].

Among the many activities that could be carried out by HSA-heme-Fe, only the (pseudo-)enzymatic activities having a biological relevance in the maintenance of anti-oxidant homeostasis in extracellular fluids are reported here [1,4,14,15,199,200]. HSA-heme-Fe facilitates RNS and ROS scavenging [4,17,200,201] and possesses weak catalase and peroxidase activities [14,142,202] similarly to sperm whale myoglobin and human hemoglobin [17,203,204].

The heme-based catalytic activities of HSA are allosterically modulated by drugs, outlining the role of heterotropic ligands in influencing HSA-heme-Fe properties [4,17,200,201]. Drugs modulate the reactivity properties of HSA-heme-Fe altering coordination state of the heme-Fe atom [17,68,205,206]. Indeed, in the active HSA-heme-Fe, the metal center is four- or five-coordinated, while it is six-coordinated in the inactive heme-protein. Drug binding to HSA-heme-Fe induces the reorientation of the Glu131-Arg145 α-helix and the axial coordination of the heme-Fe atom by His146 and Tyr161 [17,68,205,206].

Peroxynitrite scavenging by HSA-heme-Fe(II):NO is modulated by CO_2_ and abacavir [207]. Mixing HSA-heme-Fe(II)-NO and peroxynitrite solutions leads to HSA-heme-Fe(III) via the formation of the transient HSA-heme-Fe(III)-NO species [17,207]. CO_2_ increases the rate of peroxynitrite scavenging by heme-Fe(II)-NO proteins through the rapid formation of the transient reactive species CO_3_^•−^ and NO_2_^•−^ [17,208,209,210,211]. Abacavir, instead, facilitates peroxynitrite scavenging by HSA-heme-Fe(II):NO inducing allosterically the dissociation of the HSA-heme-Fe(III)-NO complex [207] (Figure 4).

Peroxynitrite scavenging by HSA-heme-Fe(III) can be modulated by CO_2_ and drugs (e.g., ibuprofen and warfarin) [66,159]. Mixing HSA-heme-Fe(III) and peroxynitrite solutions leads to HSA-heme-Fe(III) and NO_3_^−^ via the HSA-heme-Fe(III)-OONO^−^ transient [17,66,159,207]. HSA-heme-Fe(III) prevents peroxynitrite-mediated nitration of Tyr. Both in the absence and presence of CO_2_, drugs impair dose-dependently peroxynitrite isomerization by HSA-heme-Fe(III) and facilitate the nitration of Tyr. Of note, the HSA-heme-Fe(III)-catalyzed isomerization of peroxynitrite is attributed to the reactive penta-coordinated heme-Fe(III) atom. Drug binding to FA2, FA3-FA4, and FA7 impairs allosterically the peroxynitrite isomerization by HSA-heme-Fe(III) inducing the hexa-coordination of the heme-Fe(III) atom [17,66,159,207] (Figure 4). 

The reductive nitrosylation of HSA-heme-Fe(III) is in agreement with the role of the metal center to scavenge (pseudo-)enzymatically RNS and ROS [4,17,66,159,207] (Figure 4). Under anaerobic conditions, NO binding to HSA-heme-Fe(III) leads to the formation of the transient HSA-heme-Fe(III):NO, which is in equilibrium with HSA-heme-Fe(II):NO^+^. Hence, HSA-heme-Fe(II):NO^+^ undergoes nucleophilic attack by OH^−^ to yield HSA-heme-Fe(II) which in the presence of NO excess leads to the formation of the HSA-heme-Fe(II):NO adduct [4,17,66,159,207] (Figure 4). 

Lastly, HSA participates in the prevention of the toxic effects of plasma heme-Fe by trapping it [1,14,17,60,68]. This impairs the activation of the heme-Fe-atom by H_2_O_2_, the common step for both peroxidase- and catalase-like activities of heme-proteins [212] (Figure 4). HSA-heme-Fe(III) shows a weak catalase and peroxidase activities oxidizing phenolic compounds like *p*-cresol [14] and to 2,2′-azinobis(3-ethylbenzothiazoline-6-sulfonate) [202]. The main factors affecting the catalase and peroxidase activity of HSA-heme-Fe(III) are the reduced accessibility of the metal center, and the lack of a Arg residue in the HSA-heme-Fe(III) pocket that in peroxidases assists the cleavage of bound peroxide and accelerates the formation of the active species [14].

## 5. Conclusions and Perspectives 

The multiple (pseudo-)enzymatic activities of HSA and of HSA-heme-Fe are mostly based on the highly reactive residues Cys34, Lys199, and Tyr411, which are involved in the HSA anti-oxidant, esterase, and glucoronidase activities. Overall, the low (pseudo-) enzymatic activities of HSA are counterbalanced by its high concentration in plasma. 

The enzymatic properties of HSA are physiologically relevant and can affect the metabolism of different molecules (e.g., ROS, lipids, proteins, cholesterol, and drugs). Several factors can modulate HSA enzymatic functions, such as: (i) binding of allosteric effectors and competitive inhibitors; (ii) chemical modifications (e.g., glycation and oxidation); (iii) genetic inherited diseases (e.g., analbuminemia and hypoalbuminemia), (iv) pathological conditions (e.g., diabetes, liver diseases, atherosclerosis, and cancer), and (v) age. For instance, conformational changes induced upon ligand binding and/or covalent labeling may result in the allosteric modulation of HSA activity. In addition, ligands that bind to Sudlow’s site I or site II can modulate competitively the (pseudo-)enzymatic activity of HSA. 

Reactive species binding to HSA and Cys34 oxidation can alter the protein binding properties towards several ligands such as drugs and toxic substances. In turn, ligand binding to HSA can affect the reactivity of the Cys34 thiol group and can modulate HSA anti-oxidant properties. Such modulation can be used to strengthen or to weaken HSA anti-oxidant functions for the development of therapeutic strategies for diseases associated with oxidative stress.

Beside the anti-oxidant activity, HSA also plays a role in plasma detoxification from many toxicants such as organophosphorus (OP) agents (e.g., insecticides, pesticides, chemical weapons). HSA can bind OPs at both Sudlow’s sites I and II inducing their esterase-dependent cleavage. Despite the slow catalytic cleavage of OPs, HSA acts as a carrier of OPs to cholinesterase sites, where the enzymatic cleavage occurs with a high efficiency. Notably, HSA esterase activity should be considered also from the pharmaceutical and clinical points of view due to its relevant role in pro-drugs activation. Indeed, HSA has been largely employed for drug delivery and drug half-life extension. Besides, these enzymatic properties of HSA can be also exploited by chemical industries, in the so called “green chemistry” field, as a potential scaffold to produce libraries of catalysts with improved proficiency and stereo-selectivity. 

## Figures and Tables

**Figure 1 ijms-22-10086-f001:**
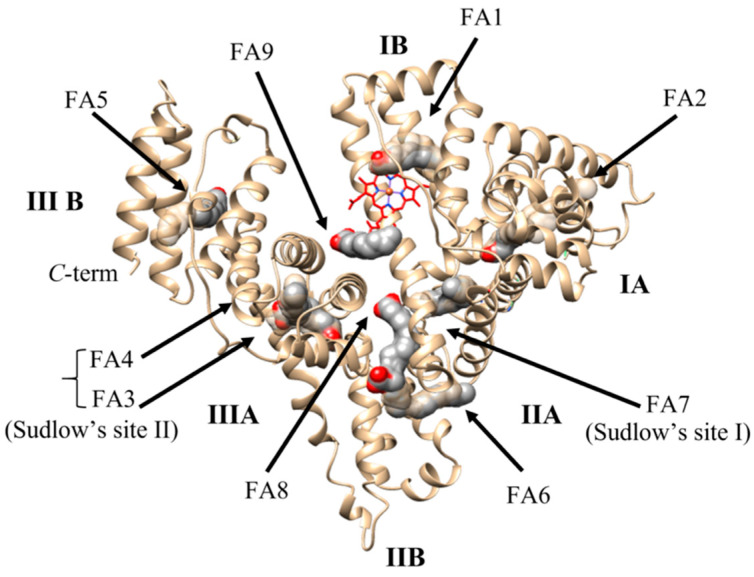
Three-dimensional structure of HSA (PDB ID: 1AO6) [40]. Fatty acids (FA) bound to specific sites are rendered as space-fill (gray). The heme is rendered as sticks (red). The picture has been drawn with UCSF-Chimera package [45].

**Figure 2 ijms-22-10086-f002:**
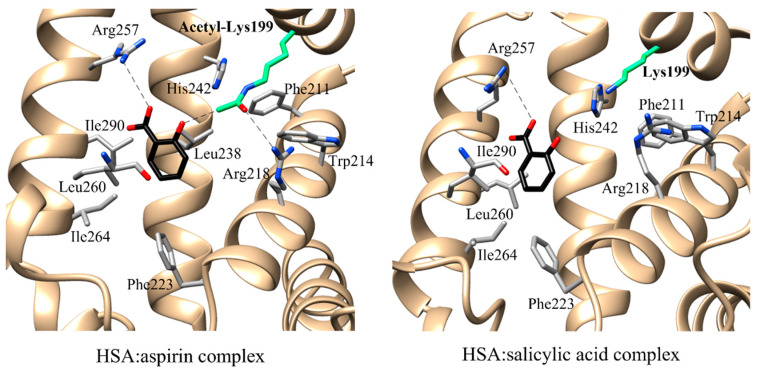
Structural environment of the HSA:aspirin complex (PDB ID: 2I2Z) [97] and of the HSA:salicylic acid adduct (PDB ID: 2I30) [97]. Acetyl-Lys199 and Lys199 residues are showed in green. Hydrogen bonds are shown in blue dash lines. Salicylic acid is rendered in sticks (black). Pictures have been drawn with UCSF-Chimera package [45].

**Figure 3 ijms-22-10086-f003:**
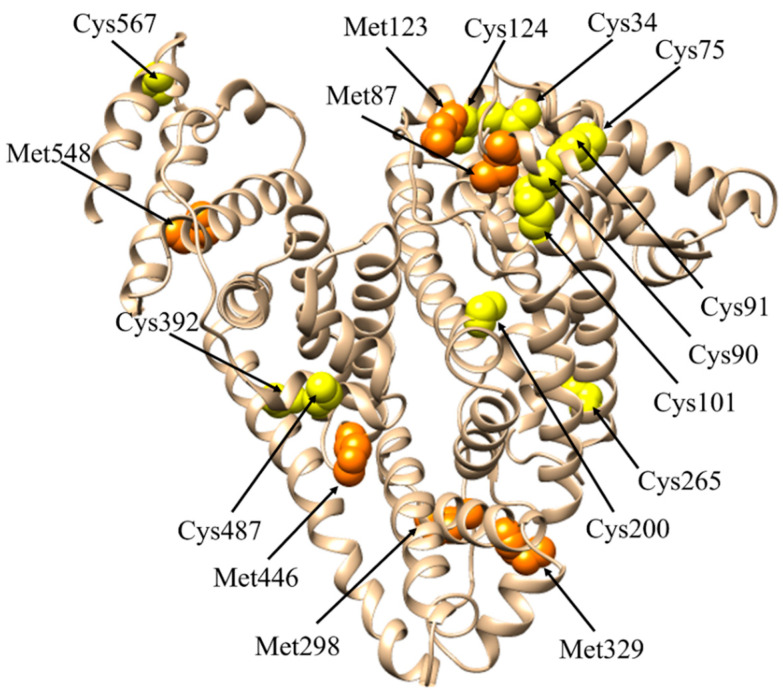
Met and Cys residues involved in HSA redox reactions. Met residues are shown as orange spheres, whereas Cys residues are depicted as yellow spheres. The picture has been drawn with UCSF-Chimera package [45].

**Figure 4 ijms-22-10086-f004:**
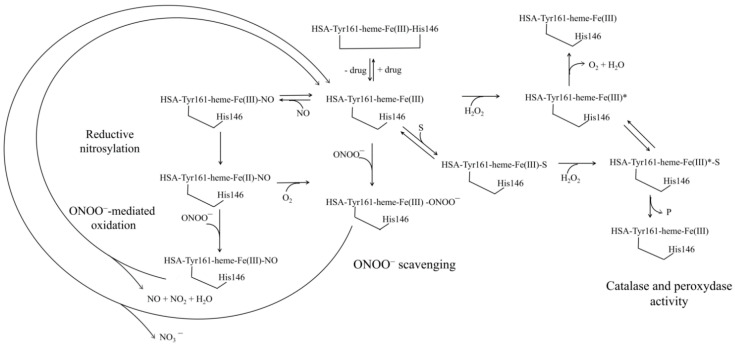
Scheme of the (pseudo-)enzymatic properties of HSA-heme-Fe. Exa-coordinated HSA-Tyr161-heme-Fe(III)-His146 indicates the inactive heme-protein whereas pentacoordinated HSA-Tyr161-heme-Fe(III) indicates the active form. HSA-Tyr-161-heme-Fe(III)* is the active HSA-Tyr-161-heme-Fe(III); S is the substrate; HSA-Tyr-161-heme-Fe(III)-S is the HSA-Tyr-161-heme-Fe(III) substrate adduct, HSA-Tyr161--heme-Fe(III)*-S is the active HSA-Tyr161-heme-Fe(III) substrate adduct.

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
