# Peer review of "Serum Albumin: A Multifaced Enzyme"

_ijms, 2021, doi:10.3390/ijms221810086_

Round 1
Reviewer 1 Report
In this manuscript, the authors reviewed HSA, focusing on various enzymatic properties of HSA. This manuscript was well summarized and beneficial. An addition of a summarized Table, and revisions of the Figures will improve the manuscript.
A Table that summerizes the relationship between various enzymatic properties and the positions (FA1 to FA9 or related amino acid residues) of HSA should be created for clarify.
[Figure 2]
The caption "Aspirin and salicylic acid are rendered in sticks (black)" seems wrong, because both ligands are salicylic acid. Difference between the two figures should be stressed (especially Acetyl-Lys199 or Lys199).
[Figure 3]
Met298, Met329, Cys75, Cys90, Cys91 should be indicated. Cys92 does not exist in the figure.
[2.1 HSA structure (Page 2)]
Because subdomains A and B of each domain have 6 and 4 alpha-helices, respectively, "that are built from 4 and 6 alpha-helices" should be changed into "that are built from 6 and 4 alpha-helices".
[3.1 HSA enzymatic properties (Page 5)]
"Ile260" should be "Leu260".
[3.5 Aldolase activity of HSA and BSA (Page 8)]
Because comparison between BSA and HSA is not the main topic of the manuscript, it is better to remove descriptions regarding BSA from the present manuscript.
Author Response
Reviewer 1
In this manuscript, the authors reviewed HSA, focusing on various enzymatic properties of HSA. This manuscript was well summarized and beneficial. An addition of a summarized Table, and revisions of the Figures will improve the manuscript.
Reviewer 1.1. Table that summarizes the relationship between various enzymatic properties and the positions (FA1 to FA9 or related amino acid residues) of HSA should be created for clarify.
Authors 1.1. Table 1 summarizing the catalytic amino acid residues of HSA has been added to the revised text.
Reviewer 1.2. [Figure 2] The caption "Aspirin and salicylic acid are rendered in sticks (black)" seems wrong, because both ligands are salicylic acid. Difference between the two figures should be stressed (especially Acetyl-Lys199 or Lys199).
Authors 1.2. We apologize for this oversight. The caption of the Figure 2 has been corrected as follows “Acetil-Lys199 and Lys199 in the HSA:aspirin complex and in the HSA:salicylic acid complex, respectively, are in green. Hydrogen bonds are blue dash lines. Salicylic acid is rendered in black sticks. Pictures have been drawn with UCSF-Chimera package [45]” (lines 266-268). In Figure 2, acetyl-Lys199 has been highlighted.
Reviewer 1.3. [Figure 3] Met298, Met329, Cys75, Cys90, Cys91 should be indicated. Cys92 does not exist in the figure.
Authors 1.3. According to the suggestion, Figure 3 has been revised.
Reviewer 1.4. [2.1 HSA structure (Page 2)] Because subdomains A and B of each domain have 6 and 4 alpha-helices, respectively, "that are built from 4 and 6 alpha-helices" should be changed into "that are built from 6 and 4 alpha-helices".
Authors 1.4. The text has been amended as required (line 72).
Reviewer 1.5. [3.1 HSA enzymatic properties (Page 5)] "Ile260" should be "Leu260".
Authors 1.5. We apologize for this oversight. Ile 260 has been replaced with Leu260 in the revised text (lines 250, 251, and 259).
Reviewer 1.6. [3.5 Aldolase activity of HSA and BSA (Page 8)] Because comparison between BSA and HSA is not the main topic of the manuscript, it is better to remove descriptions regarding BSA from the present manuscript.
Authors 1.7. All statements referring to BSA have been removed from the paper. Therefore, paragraph 3.5, refers only to human SA.
Reviewer 2 Report
The Authors presented the paper "Serum albumin: a multifaced enzyme".
Specific comments
1) The Idea to present the enzymatic properties of albumin is interesting and fresh. However, the presented in the review information is limited. I recommend enlarging this part of the review, to present the possible mechanisms as a picture or smth like this, properties of enzymatic activities, etc.
2) 2.2. Ligand binding properties of HSA
I think in this par. you can present the Table or scheme to present the whole possible molecules which can be bind by albumin. In the review, "Human serum albumin: From bench to bedside" presented much more information about it.
3) Also, I haven't found enough information as was mentioned in the abstract about:
Catalytic properties of HSA are modulated by allosteric effectors, competitive inhibitors, chemical modifications, genetic inherited diseases, pathological conditions, and age.
It will be very good to do a new subsection about such interesting allosteric regulation because of the cys-34 modification, ligand binding, etc. if it is possible. It is a very interesting area of albumin functions.
4) As you know the albumin present in the organism is a monomer and dimer, and oligomer fraction. Also, I haven't seen any information about dimers, oligomers, and their functions in the relation to the disease or enzymatic activity. It was obtained some noncovalent dimers of albumin which can alter its biological properties (Molecules 2021).
5) Maybe you can present some information about ros formation, functions, amyloid-formation of N-homocysteinylated albumin. It is one of the main posttranslational modifications of albumin. But in the review, there is no information about it at all. But it is in your consideration.
Author Response
Reviewer 2
Reviewer 2.1. Specific comments. The Idea to present the enzymatic properties of albumin is interesting and fresh. However, the presented in the review information is limited. I recommend enlarging this part of the review, to present the possible mechanisms as a picture or smth like this, properties of enzymatic activities, etc.
Authors 2.1. According to the Reviewer’s suggestion, the modulation of HSA enzymatic activities has been analyzed in the revised text. In particular, the revised manuscript now reports: (i) a table summarizing the substrates and co-factors of each enzymatic process, as well as the catalytic sites and the catalytic amino acids involved (Table 1); (ii) the role of HSA oligomerization in the modulation of HSA enzymatic activities (paragraph 2.2); and (iii) the effects of post-translational modifications (i.e., homocysteinylation, oxidation, and glycation) in the modulation of HSA enzymatic activities (paragraph 3.7.1 and subparagraphs therein).
Reviewer 2.2. Ligand binding properties of HSA. I think in this par. you can present the Table or scheme to present the whole possible molecules which can be bind by albumin. In the review, "Human serum albumin: From bench to bedside" presented much more information about it.
Authors 2.2. According to the Reviewer’s suggestion, a new Table S1 summarizing the three-dimensional structures of HSA-ligand complexes has been added to the Supplementary Materials of the revised manuscript.
Reviewer 2.3. Also, I haven't found enough information as was mentioned in the abstract about: Catalytic properties of HSA are modulated by allosteric effectors, competitive inhibitors, chemical modifications, genetic inherited diseases, pathological conditions, and age. It will be very good to do a new subsection about such interesting allosteric regulation because of the Cys-34 modification, ligand binding, etc. if it is possible. It is a very interesting area of albumin functions.
Authors 2.3. The modulation of HSA catalytic activity has been further analyzed in the revised manuscript. In particular, the modulation of HSA anti-oxidant activity by the Cys34 residue by chemical modification, glycation, fatty acids binding, and aging have been reported in the new paragraph 3.7.1 (lines 478-566). The relationship between COVID-19 and HSA oxidation has been reported (lines 511-518). The modulation of all the other catalytic activities of HSA has been reported and/or clarified throughout the revised text (lines 100-102; 278-292).
Reviewer 2.4. As you know the albumin present in the organism is a monomer and dimer, and oligomer fraction. Also, I haven't seen any information about dimers, oligomers, and their functions in the relation to the disease or enzymatic activity. It was obtained some noncovalent dimers of albumin which can alter its biological properties (Molecules 2021).
Authors 2.4. According to the Reviewer’s suggestion, HSA oligomerization and its effect on HSA catalytic activities and anti-oxidant function has been reported in the revised manuscript (lines 82-117).
Reviewer 2.5. May be you can present some information about ros formation, functions, amyloid-formation of N-homocysteinylated albumin. It is one of the main post-translational modifications of albumin. But in the review, there is no information about it at all. But it is in your consideration.
Authors 2.5. According to the Reviewer’s suggestion, a brief overview on reactive oxygen and nitrogen species formation and metabolism has been added at the beginning of paragraph 3.7 (lines 381-388). Moreover, the effects of post-translational modifications on HSA enzymatic activities have been reported in the revised text. In particular, the effects of HSA homocysteinylation and oxidation (paragraph 3.7.7.1, lines 478-523) and glycation (paragraph 3.7.7.1, lines 525-536) have been reported. Finally, the effects of HSA oligomerization on the anti-oxidant activity have been discussed (paragraph 2.2, lines 82-116).
Reviewer 3 Report
The review on enzymatic properties of human serum albumin is timely and very informative. In order to enhance the readability, this referee suggests the authors to prepare a Table(s) summarizing type of enzymatic reaction, enzymatic or pseudo-enzymatic, catalytic sites (and residues), substrates, kinetic parameters (if any), and appropriate references.
Author Response
Reviewer 3
Reviewer 3.1. The review on enzymatic properties of human serum albumin is timely and very informative. In order to enhance the readability, this referee suggests the authors to prepare a Table(s) summarizing type of enzymatic reaction, enzymatic or pseudo-enzymatic, catalytic sites (and residues), substrates, kinetic parameters (if any), and appropriate references.
Authors 3.1. According to the Reviewer’s suggestion, the Table 1 summarizing the type of enzymatic reaction, substrates, catalytic sites and residues, kinetic parameters (if any), and appropriate references has been added to the revised text.
Round 2
Reviewer 2 Report
Thank you for the high-quality and hard work. I will use this review for my further research work.